# Predictors of Assessment of Spondyloarthritis International Society (ASAS) Health Index in Axial Spondyloarthritis and Comparison of ASAS Health Index between Ankylosing Spondylitis and Nonradiographic Axial Spondyloarthritis: Data from the Catholic Axial Spondyloarthritis COhort (CASCO)

**DOI:** 10.3390/jcm8040467

**Published:** 2019-04-05

**Authors:** Hong Ki Min, Jennifer Lee, Ji Hyeon Ju, Sung-Hwan Park, Seung-Ki Kwok

**Affiliations:** 1Division of Rheumatology, Department of Internal Medicine, Konkuk University Medical Centre, Konkuk University School of Medicine, 120-1, Neungdong-ro, Gwangjin-gu, Seoul 05030, Korea; alsghdrl1921@naver.com; 2Division of Rheumatology, Department of Internal Medicine, Seoul St. Mary’s Hospital, College of Medicine, The Catholic University of Korea, 222 Banpo-daero, Seocho-gu, Seoul 06591, Korea; poohish@naver.com (J.L.); juji@catholic.ac.kr (J.H.J.); rapark@catholic.ac.kr (S.-H.P.)

**Keywords:** axial spondyloarthritis, ASAS health index, mSASSS, socioeconomic status, alcohol

## Abstract

The Assessment of Spondyloarthritis International Society (ASAS) health index (HI) is a novel tool for approaching disability, health, and functioning in spondyloarthritis (SpA). In the present study we compared ASAS HI between patients with ankylosing spondylitis (AS) and those with nonradiographic axial SpA (nr-axSpA). In addition, we identified predictors of ASAS HI. We designed this cross-sectional study using data from the Catholic Axial Spondyloarthritis COhort (CASCO), a prospective cohort from a single tertiary hospital. We compared baseline characteristics, including ASAS HI, between AS and nr-axSpA, and determined the frequency of each item constituting the ASAS HI. We used linear regression analysis to identify factors associated with ASAS HI. Total of 357 patients with axSpA—261 with AS and 96 with nr-axSpA—were included in analysis. AS patients were older and had higher ASAS HI than nr-axSpA. Among ASAS HI items, pain (item No. 1) and energy/drive (item No. 5) were the most common areas for which axSpA patients experienced discomfort. ASAS HI correlated with other SpA-related parameters such as BASDAI, ASDAS, and BASFI. Multivariable regression analysis of the axSpA group showed that high NSAID intake and mSASSS were positively associated with ASAS HI, whereas higher economic status and alcohol consumption were negatively associated with ASAS HI. Results were consistent in the AS group on subgroup analysis, whereas alcohol consumption was the only factor significantly associated with ASAS HI in the nr-axSpA group. In the present cohort study, patients with AS had poorer health status (higher ASAS HI) than those with nr-axSpA. Items proposed by AS patients (items No. 1 and 5) were the most frequently checked areas as axSpA patients feel discomfort, and this support that ASAS HI could practically assess actual discomfort of axSpA patient. ASAS HI was well correlated with known disease parameters, including activity, function, and quality of life; therefore, ASAS HI could be used in the future to represent the health status of SpA in a systematic way. Spinal structural damage (higher mSASSS), high NSAID intake, alcohol consumption, and economic status were predictors of ASAS HI in patients with axSpA, especially those with AS.

## 1. Introduction

The Assessment of Spondyloarthritis International Society (ASAS) health index (HI) was recently developed as a means of assessing the health, functioning, and disability of patients with spondyloarthritis (SpA) in an inclusive way [1]. Rheumatologists, patients with ankylosing spondylitis (AS), and other healthcare professionals participated in selecting the items comprising the ASAS HI, which were chosen from a core set of the International Classification of Functioning, Disability and Health (ICF) [1,2]. Although, several patient-reported outcomes (PROs) are available for assessing SpA with regard to activity, function, and health-related quality of life (HRQoL), these are insufficient for estimating health status of patients with SpA in a systematic way.

Items suitable for AS and representative of a wide spectrum of health and functioning indices related to AS were selected from the ICF core set to comprise the ASAS HI [1,2]. A total of 17 dichotomous items pertaining to pain, emotional functioning, sexual functioning, mobility, self-care, and even community life were ultimately selected [1]. ASAS HI was validated internationally in 23 countries including patients with axial and peripheral SpA and has been translated into 18 languages, including Korean [3,4]. The ASAS HI questionnaire is easy to complete by choosing either “I agree” or “I do not agree”, and it has high internal consistency [4].

Usually AS is considered an advanced form of axSpA, and some nonradiographic axSpA (nr-axSpA) forms can progress to AS; ~26% over 15 years of follow-up [5]. Several studies have compared the features of AS and nr-axSpA; however, ASAS HI has not yet been compared in a large cohort study. Furthermore, predictors of ASAS HI have not been identified.

The objectives of the present study were to assess ASAS HI in a large cohort of Korean patients and compare ASAS HI between AS patients and nr-axSpA patients. In addition, we aimed to identify the most commonly checked “I agree” item among the 17 ASAS HI items and identify predictors of ASAS HI.

## 2. Patients and Methods

### 2.1. Patients

AxSpA patients were recruited from a single tertiary referral hospital, Seoul Saint Mary’s Hospital, and enrolled in a prospective cohort: the Catholic Axial Spondyloarthritis COhort (CASCO). Inclusion criteria: (1) fulfilling either modified New York (mNY) criteria for AS or ASAS classification criteria for axSpA [6,7] and (2) age over 18. The study was conducted in accordance with the Declaration of Helsinki (1964). Written informed consent was obtained from each patient before they were enrolled. This study was approved by the Institutional Review Board of Seoul St. Mary’s Hospital (KC15OISI0012) at 29 December 2014.

### 2.2. Study Design

This was a cross-sectional study. Patients were divided into AS and nr-axSpA groups and then baseline characteristics were compared between them.

### 2.3. Collected Data

Baseline characteristics, including demographic, laboratory, and radiographic data, and socioeconomic status, were collected at enrollment. Laboratory data included erythrocyte sedimentation rate (ESR), C-reactive protein (CRP), and human leukocyte antigen (HLA)-B27. Elevated CRP was defined as a CRP level greater than 0.5 mg/dL. Lateral views of the cervical and lumbar spine were taken to calculate the modified stoke ankylosing spondylitis spinal score (mSASSS), and plain radiographs of the pelvis were used to evaluate the grade of sacroiliitis. Sacroiliitis was measured according to the mNY criteria and assigned to one of five grades (grade 0, normal; grade 1, suspicious change; grade 2, localized areas with sclerosis or erosion without alterations in joint width; grade 3, moderate or advanced sacroiliitis with one or more of the following: erosions, sclerosis, widening, narrowing, or partial ankylosis; grade 4, total ankyloses) [6]. Sacroiliitis grade and mSASSS were assigned by an experienced rheumatologist, Dr. Min, and the radiographs were provided in the Digital Imaging and Communications in Medicine format after erasing the patients’ information. The radiologic scoring was made by using radiographs of pelvis and spine filmed at the time of enrollment to CASCO. The sum of sacroiliitis grade was calculated by adding the grade of right and left sides. Syndesmophyte was counted from the lower border of the second cervical spine (C2) to the upper border of the first thoracic spine (T1), and from the lower border of T12 to the upper border of sacrum.

Participating patients completed paper questionnaires assessing PROs: ASAS HI, environmental factors related to ASAS HI [1], the Bath Ankylosing Spondylitis Disease Index (BASDAI) [8], the Bath Ankylosing Spondylitis Functioning Index (BASFI) [9], the EuroQol five dimensions questionnaire (EQ-5D), the EQ-visual analogue scale (VAS) [10], Ankylosing Spondylitis Disease Activity Score (ASDAS) [11], patient global assessment (PGA), and overall spinal pain and nocturnal spinal pain VAS. EQ-5D was converted into a ‘time trade-off’ (TTO) value based on previous reference data from the Republic of Korea [12]. ASAS HI ranged from 0 to 17, with a higher score indicating inferior health status. In case of missing data, the total ASAS HI score was calculated by (item summation score/17-number of missing items) × 17. When more than three items were missing, the total ASAS HI score was not allocated and excluded from the analysis. The cut-off distinguishing between good and moderate ASAS HI was 5, and between moderate and poor was 12. BASDAI, ASDAS, BASFI, and PGA ranged from 0 to 10, and a higher score indicated worse disease activity, physical functioning, and self-assessment. Higher EQ-5D TTO and EQ-VAS meant better QoL.

Socioeconomic status was assessed with regard to salary, education level, and marital status. Marital status was categorized as married or single/divorced/bereaved. Smoking and alcohol consumption were grouped as currently smoking/consuming alcohol or not. Body mass index (BMI) was measured at the time of enrollment, and obesity was defined as a BMI greater than 25 kg/m^2^ based on Korean obesity guidelines [13]. Current medications were confirmed by reviewing the patient electronic medical record (EMR) and asking each patient the actual amount of medication taken. The ASAS nonsteroidal anti-inflammatory drug (NSAID) index was calculated, and based on data from a previous study, an NSAID index greater than 50 corresponded to high NSAID intake [14,15]. Patients checked “yes” if they experienced or were diagnosed with features of SpA including uveitis, dactylitis, psoriasis, and inflammatory bowel disease (IBS) at least once.

### 2.4. Statistical Analysis

Continuous variables were compared using Student’s *t*-test, and results were presented as the mean ± standard deviation (SD). Categorical variables were compared using the chi squared test or Fisher’s exact test. Spearman correlation coefficients were calculated to assess the correlation between ASAS HI and PROs. Candidate variables for regression analysis were referred to the previous studies which investigated associated factors of HRQoL and structural progression in SpA [16,17]. Linear regression analysis was performed to determine predictors of ASAS HI. Factors achieving *p* < 0.20 on univariable regression analysis were included in the multivariable regression analysis model by backward stepwise regression. Linear regression analysis was performed in the overall axSpA population, then in the AS and nr-axSpA subgroups. *p*-values < 0.05 were considered statistically significant. All tests were performed using the R software (R for Windows 3.3.2; The R Foundation for Statistical Computing, Vienna, Austria).

## 3. Results

### 3.1. Baseline Characteristics of Patients with Axial Spodyloarthritis (axSpA) and Comparison between Those with Ankylosing Spondylitis (AS) and Nonradiographic (nr) axSpA

A total of 372 patients with axSpA were enrolled into CASCO from January 2015 to April 2017. Among them, 15 did not complete the paper questionnaire and were excluded from our analysis. Data for a total of 357 patients with axSpA—261 with AS and 96 with nr-axSpA—were analyzed (Table 1). All patients had history of inflammatory back pain. Patients in the AS group were older and had longer disease duration. As expected based on the fact that patients with AS were older, other features, including salary and marital status, were significantly different between the nr-axSpA and AS groups. Furthermore, radiographic spinal structural damage (mSASSS) was more severe in the AS group. ASAS HI and environmental factors related to ASAS HI were higher in the AS group than in the nr-axSpA group (3.8 ± 3.5 vs. 2.7 ± 2.8, *p* = 0.003; 2.2 ± 1.7 vs. 1.8 ± 1.2, *p* = 0.009, respectively). Health status, determined based on ASAS HI, showed that most patients with axSpA were in good health (75.9% in good health, Table 2). The item most frequently checked as “I agree” was item No. 1 (57.7% in AS, 44.8% in nr-axSpA), and the second most frequent was item No. 5 (51.2% in AS, 44.8% in nr-axSpA). Item No. 1 represented health status for “pain; Pain sometimes disrupts my normal activities” and item No. 5 for “energy and drive; I am often exhausted”. Except for items No. 1 and 3, items representing “pain” and “moving around; I have problems running”, there were no significant differences between the AS and nr-axSpA groups (Figure 1). A copy of the ASAS HI is supplied as Appendix A.

### 3.2. Correlation between ASAS HI and PROs

EQ-5D-TTO and BASFI were highly correlated with the ASAS HI (EQ-5D-TTO, Rho = −0.71, *p* < 0.001, and BASFI, Rho = 0.65, *p* < 0.001). In addition, other PROs, such as BADAI, ASDAS, PGA, and physician global assessment (PhyGA), were significantly positively correlated with ASAS HI (Table 3).

### 3.3. Predictors of ASAS HI in axSpA, AS, and nr-axSpA

We performed linear regression analysis to identify predictors of ASAS HI in patients with total axSpA, including AS and nr-axSpA. Univariable regression analysis showed that age, high NSAID intake, sum of sacroiliitis grade, syndesmophyte presence, and mSASSS had significant positive associations with ASAS HI. Higher education level, higher economic status, and alcohol consumption were negatively associated with ASAS HI. On multivariable regression analysis, higher economic status and alcohol consumption were negatively associated with ASAS HI (β = −1.055, *p* = 0.013; β = −1.126, *p* = 0.004, respectively). High NSAID intake and mSASSS had significant positive associations with ASAS HI (β = 1.050, *p* = 0.003; β = 0.035, *p* = 0.006, respectively; Table 3). Subgroup analysis of AS group had similar results to those for the overall axSpA group except for HLA-B27 (Table 4). In the nr-axSpA group only alcohol consumption was negatively associated with ASAS HI (Table 4).

## 4. Discussion

The present study compared several PROs, including ASAS HI between patients, with AS and nr-axSpA. In addition, we identified the frequency of each item on the ASAS HI questionnaire in Korean patients with axSpA. Similar to previous studies, we found correlations between ASAS HI and existing axSpA disease parameters including BASDAI, ASDAS, and BASFI [3,4,18]. Finally, for the first time, predictors of ASAS HI were identified in patients with axSpA and subgroups of patients with AS and nr-axSpA.

The impact of axSpA on quality of life is considerable in various respects. Although existing PROs can characterize disease activity, physical functioning, and HRQoL, they are not suitable for determining the overall health and disability status of patients with axSpA. The biopsychosocial model proposed by ICF was used to develop the basis of the ASAS HI [2]. The ASAS HI items were selected from a core set of ICF variables, and rheumatologists, health care experts, and even patients with AS were involved in the process of item selection. Several items were proposed by patients with AS, and items covering self-care, participation in community life, and leisure activities that were important from the patient perspective were included in the ASAS HI. We evaluated the frequency of items checked as “I agree”, and items No. 1 and 5 were most commonly checked. These items were proposed by patients with AS patient [1], and this supports the notion that the ASAS HI is a practical assessment of a patient’s actual discomfort in daily life.

Previous studies have already showed that there is a significant correlation between ASAS HI and several PROs including BASDAI, BASFI, ASDAS, and EQ-VAS [3,4,18]. Similar results were observed in the present study, which supports the notion that known PROs in SpA have similar features to the items on the ASAS HI questionnaire, and ASAS HI is a comprehensive assessment of health status. Further studies using the ASAS HI as an assessment instrument to evaluate the effect of novel therapy in SpA might increase the reliability of ASAS HI in clinical practice.

ASAS HI was developed recently; therefore, the predictors of ASAS HI had not been demonstrated until now. In the present study, we determined ASAS HI prospectively in a single cohort and identified several factors that are associated with ASAS HI. Higher economic status and alcohol consumption were negatively associated with ASAS HI, whereas high NSAID intake and higher mSASSS were positively associated with ASAS HI. The associations between socioeconomic status and SpA disease activity, physical functioning, and HRQoL have been evaluated in limited studies. Roussou E. et al. showed that patients with AS from a higher occupational group had lower pain, depression, and disease activity than patients from a lower occupational group [19]. Another study revealed that physical HRQoL was positively associated with educational level and employment status, but mental HRQoL did not have such an association [20]. In the present study, higher educational level and economic status were significantly associated with lower ASAS HI on univariable regression analysis. Higher economic status remained significant on multivariable regression analysis. The aforementioned results support the notion that overall health, functioning, and disability in axSpA are not fully explained by the disease itself but are also affected by socioeconomic status. This revelation might aid physicians in understanding that actual discomfort due to axSpA may not arise entirely from the disease itself but could also be affected by baseline socioeconomic status.

Patients with axSpA who consumed alcohol had lower BASDAI, ASDAS, and BASFI in a cross-sectional study [21]. Another study of AS showed that alcohol consumption tends to increase BASDAI [22]. We consistently found that alcohol consumption was negatively associated with ASAS HI. Further studies including a larger sample size and long-term data are required to determine the overall effect of alcohol consumption on health, functioning, and disability in patients with axSpA.

Several studies showed advanced spinal structural damage was associated with poor PROs outcomes in patients with AS. Two prior studies found a positive correlation between mSASSS and BASFI [23,24]. Spinal mobility assessed by the Bath Ankylosing Spondylitis Metrology Index (BASMI) was significantly correlated with mSASSS [25]. In the present study, we found that ASAS HI was positively associated with mSASSS, especially in patients with AS. This result is consistent with those of previous studies that assessed the correlation between mSASSS and physical functioning (BASFI) or spinal mobility (BASMI). There was not a significant association between ASAS HI and mSASSS in the nr-axSpA group. The aforementioned discrepancy between AS and nr-axSpA might arise from our small sample size of nr-axSpA (n = 96) and the low mSASSS in the nr-axSpA group. High NSAID intake was also positively associated with ASAS HI. The ASAS NSAID index may represent the actual dosage of NSAID by assessing the daily NSAID dose and the weekly frequency [14]. Recent treatment guidelines from the ASAS/European League Against Rheumatism (EULAR) recommend NSAIDs as first-line therapy, and consider risks and benefits individually [26]. High NSAID intake might simply mean high disease burden, and patients with axSpA may have good adherence to NSAID treatment because of uncontrolled pain or stiffness. In addition, the positive association between high NSAID intake and ASAS HI supports the notion that patients with uncontrolled pain or stiffness may have poor health status.

Comparison between HLA-B27 positive SpA and negative SpA has been studied in several studies; however, revealing the association between HLA-B27 positivity and clinical parameters was rare. A study from Spain showed HLA-B27 negative AS patients presented higher BASDAI and BASFI than HLA-B27 positive AS patients [27]. In the present study, ASAS HI was negatively associated with HLA-B27 positivity in AS patients. Although clinical parameters were different in present and previous study, the results of the two studies were similar. In addition, the regression analysis of total axSpA and nr-axSpA did not show a significant association between HLA-B27 positivity and ASAS HI, and these might come from less HLA-B27 positive patients were included in total axSpA and nr-axSpA group than AS group. To clarify the role of HLA-B27 on ASAS HI, further studies with larger sample size are needed.

The present study had some limitations. First, the nr-axSpA patients included in present study was relatively small. Patients with AS had similar results to the overall axSpA group on regression analysis, whereas patients with nr-axSpA did not. This result may be due to the relatively small size of the nr-axSpA group in the present study. However, CASCO patients were enrolled at a single tertiary hospital, therefore data collection was standardized, which increased the reliability of the collected data. Second, the study design was cross-sectional, therefore we were not able to assess the long-term impact of several factors proven to be associated with ASAS HI in this study. The CASCO is a prospective cohort, therefore extended research can be done in the future.

We found that ASAS HI was higher in patients with AS than in those with nr-axSpA, and we identified the common ASAS HI items that were actually related to discomfort in patients with axSpA. Furthermore, this is the first study to identify predictors of ASAS HI in patients with axSpA.

## Figures and Tables

**Figure 1 jcm-08-00467-f001:**
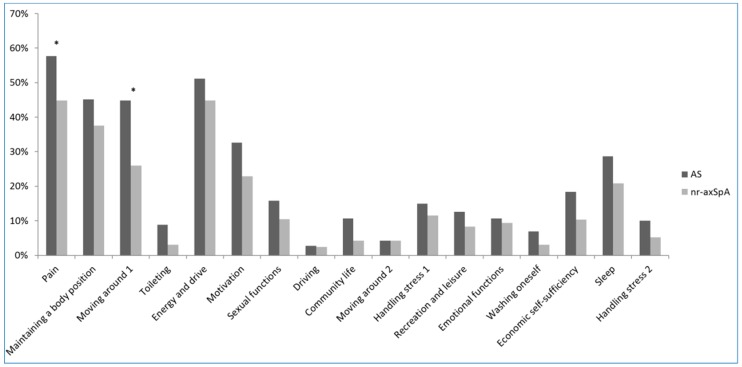
The frequency of ASAS HI items checked as “I agree” in the AS and nr-axSpA groups. * *p* < 0.05.

**Table 1 jcm-08-00467-t001:** Baseline demographic characteristics and comparison between nonradiographic axial SpA (nr-axSpA) and AS.

Characteristics	Total axSpA (N = 357)	Nr-axSpA (N = 96)	AS (N = 261)	*p* ^†^
Age (years)	38.7 ± 11.2	34.8 ± 10.7	40.1 ± 11.1	**<0.001**
Diagnosed age (years)	31.0 ± 11.5	29.3 ± 10.9	31.6 ± 11.6	0.083
Disease duration (years)	7.6 ± 6.6	5.3 ± 3.8	8.4 ± 7.2	**<0.001**
Male gender (N, %)	273 (76.5%)	66 (68.8%)	207 (79.3%)	0.052
BMI (kg/m^2^)	24.0 ± 3.3	23.3 ± 3.2	24.3 ± 3.2	**0.008**
Education				0.864
Below high school	102 (29.8%)	26 (28.6%)	76 (30.3%)	
College or postgraduate	240 (70.2%)	65 (71.4%)	175 (69.7%)	
Salary				**0.026**
<50,000 US dollar/year	244 (71.8%)	74 (81.3%)	170 (68.3%)	
≥50,000 US dollar/year	96 (28.2%)	17 (18.7%)	79 (31.7%)	
Marriage				**0.001**
Single/divorced/bereaved	151 (42.4%)	55 (57.3%)	96 (36.9%)	
Married	205 (57.6%)	41 (42.7%)	164 (63.1%)	
Current smoker	99 (28.0%)	17 (17.9%)	82 (31.8%)	**0.015**
Current alcohol drinker	244 (69.1%)	68 (70.8%)	176 (68.5%)	0.767
Uveitis	161 (45.5%)	33 (34.4%)	128 (49.6%)	**0.015**
IBD	6 (1.7%)	1 (1.0%)	5 (1.9%)	0.903
Dactylitis	28 (7.9%)	10 (10.4%)	18 (7.0%)	0.398
Psoriasis	17 (4.8%)	3 (3.1%)	14 (5.4%)	0.535

AS—ankylosing spondylitis; axSpA—axial spondyloarthritis; BMI—body mass index; IBD—inflammatory bowel disease; nr-axSpA—nonradiographic axial spondyloarthritis; Data are shown as mean ± standard deviation (SD). ^†^ Comparison between AS and nr-axSpA.

**Table 2 jcm-08-00467-t002:** Comparison of laboratory, radiographic, and axSpA related parameters between nr-axSpA and ankylosing spondylitis (AS).

Variables	Total axSpA (N = 357)	Nr-axSpA (N = 96)	AS (N = 261)	*p* ^†^
CRP elevation (>0.5 mg/dL)	67 (18.8%)	8 (8.3%)	59 (22.7%)	**0.003**
HLA-B27 positive	310 (93.1%)	82 (87.2%)	228 (95.4%)	**0.016**
High NSAID intake (ASAS NSAID index ≥ 50)	195 (54.6%)	43 (44.8%)	152 (58.2%)	**0.032**
Sulfasalazine	122 (34.3%)	32 (33.3%)	90 (34.6%)	0.920
TNF-α inhibitor	170 (47.8%)	41 (42.7%)	129 (49.6%)	0.299
Sum of sacroiliitis grade (0–8)	5.0 ± 2.1	2.1 ± 0.9	6.0 ± 1.4	**<0.001**
mSASSS (0–72)	12.2 ± 18.7	3.0 ± 5.1	15.6 ± 20.6	**<0.001**
Presence of syndesmophyte	210 (58.8%)	36 (37.5%)	174 (66.7%)	**<0.001**
BASDAI (0–10)	3.1 ± 1.9	2.7 ± 1.7	3.2 ± 2.0	**0.023**
ASDAS-CRP (0–10)	1.9 ± 0.9	1.6 ± 0.7	2.0 ± 0.9	**<0.001**
ASDAS-ESR (0–10)	2.0 ± 0.9	1.7 ± 0.7	2.1 ± 1.0	**<0.001**
BASFI (0–10)	1.0 ± 1.4	0.5 ± 0.8	1.1 ± 1.5	**<0.001**
PGA (0–10)	3.2 ± 2.2	2.6 ± 1.9	3.4 ± 2.3	**0.001**
Spinal pain VAS (0–10)	2.8 ± 2.4	2.2 ± 2.0	3.1 ± 2.5	**0.001**
Nocturnal spinal pain VAS (0–10)	2.3 ± 2.3	1.7 ± 2.1	2.5 ± 2.3	**0.003**
PhyGA (0–10)	2.4 ± 1.6	2.1 ± 1.6	2.4 ± 1.6	0.131
EQ-5D-TTO (0–1)	0.79 ± 0.10	0.82 ± 0.08	0.78 ± 0.10	**0.001**
EQ-VAS (0–100)	72.0 ± 18.1	72.9 ± 18.5	71.7 ± 17.9	0.577
ASAS health index (0–17)	3.5 ± 3.4	2.7 ± 2.8	3.8 ± 3.5	**0.003**
Health index				0.053 *
Poor (≥12)	13 (3.6%)	1 (1.0%)	12 (4.6%)	
Moderate (5–12)	73 (20.4%)	14 (14.6%)	59 (22.6%)	
Good (≤5)	271 (75.9%)	81 (84.4%)	190 (72.8%)	
Environmental factor related to ASAS health index (0–9)	2.1 ± 1.6	1.8 ± 1.2	2.2 ± 1.7	**0.009**

AS—ankylosing spondylitis; ASAS—Assessment of Spondyloarthritis International Society; ASDAS—Ankylosing Spondylitis Disease Activity Score; axSpA—axial spondyloarthritis; BASDAI—Bath Ankylosing Spondylitis Disease Activity Index; BASFI—Bath Ankylosing Spondylitis Functional Index; CRP—C-reactive protein; EQ-5D—EuroQol-5 dimensions; ESR—erythrocyte sedimentation rate; HLA—human leukocyte antigen; modified Stoke Ankylosing Spondylitis Spinal Score; nr-axSpA—nonradiographic axial spondyloarthritis; NSAID—nonsteroidal anti-inflammatory drug; PGA—patient global assessment; PhyGA—physician global assessment; TNF—tumor necrosis factor; TTO—time trade-off; VAS—visual analogue scale; Data are shown as mean ± standard deviation (SD). ^†^ Comparison between AS and nr-axSpA. * The Mantel–Haenszel χ2 test was used.

**Table 3 jcm-08-00467-t003:** Spearman correlation between ASAS HI scores and other patient-reported outcomes (PROs).

	Rho	*p*
BASDAI	0.58	<0.001
ASDAS-CRP	0.56	<0.001
ASDAS-ESR	0.52	<0.001
BASFI	0.65	<0.001
EQ-5D-TTO	−0.71	<0.001
EQ-VAS	−0.54	<0.001
PGA	0.53	<0.001
PhyGA	0.49	<0.001

**Table 4 jcm-08-00467-t004:** Univariable and multivariable linear regression analysis of predicting ASAS HI in total axSpA, AS, and nr-axSpA.

Variables	Total axSpA						AS						Nr-axSpA		
	Univariable			Multivariable			Univariable			Multivariable			Univariable		
	β	SE	*p*	β	SE	*p*	β	SE	*p*	β	SE	*p*	β	SE	*p*
Age (year)	0.076	0.015	<0.001	0.035	0.020	0.079	0.094	0.019	<0.001	0.046	0.024	0.057	0.001	0.026	0.974
Male gender	−0.699	0.417	0.095	−0.822	0.437	0.061	−1.083	0.533	0.043	−1.065	0.554	0.056	−0.318	0.609	0.603
Obesity (BMI ≥ 25 kg/m^2^)	0.064	0.372	0.863				−0.059	0.448	0.896				−0.008	0.636	0.989
Higher education level (college or postgraduate)	−1.192	0.397	0.003				−1.394	0.481	0.004				−0.555	0.650	0.396
Higher economic status (≥50,000 US dollar/year)	−0.892	0.407	0.029	−1.045	0.423	0.014	−1.148	0.480	0.018	−1.205	0.487	0.014	−0.617	0.753	0.415
Married	−0.180	0.360	0.617				−0.495	0.451	0.274				−0.050	0.571	0.931
Current smoker	0.141	0.398	0.723				0.119	0.470	0.800				−0.480	0.743	0.518
Current alcohol drinker	−1.437	0.381	<0.001	−1.045	0.396	0.009	−1.421	0.466	0.003	−1.052	0.474	0.028	−1.395	0.605	0.023
High NSAID intake	1.169	0.351	0.001	1.034	0.357	0.004	1.483	0.431	0.001	1.419	0.433	0.001	−0.038	0.568	0.947
TNF-α blocker user	0.435	0.356	0.223				0.450	0.436	0.303				0.177	0.571	0.758
HLA-B27 positive	−1.201	0.707	0.090	−1.140	0.720	0.115	−2.694	1.047	0.011	−2.113	1.008	0.037	−0.228	0.861	0.792
Sum of sacroiliitis grade	0.255	0.082	0.002				0.247	0.155	0.113				−0.040	0.302	0.895
Existence of syndesmophyte	1.273	0.355	<0.001				1.261	0.455	0.006				0.651	0.580	0.264
mSASSS	0.052	0.009	<0.001	0.035	0.013	0.006	0.048	0.010	<0.001	0.032	0.014	0.021	0.039	0.056	0.485

ASDAS—Ankylosing Spondylitis Disease Activity Score; BASDAI—Bath Ankylosing Spondylitis Disease Activity Index; BMI—body mass index; HLA—human leukocyte antigen; mSASSS—modified Stoke Ankylosing Spondylitis Spinal Score; NSAID—nonsteroidal anti-inflammatory drug; SE—standard error; TNF—tumor necrosis factor.

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
