# Peer review of "Predictors of Assessment of Spondyloarthritis International Society (ASAS) Health Index in Axial Spondyloarthritis and Comparison of ASAS Health Index between Ankylosing Spondylitis and Nonradiographic Axial Spondyloarthritis: Data from the Catholic Axial Spondyloarthritis COhort (CASCO)"

_jcm, 2019, doi:10.3390/jcm8040467_

Round 1
Reviewer 1 Report
The authors described the ASAS health index in cohort of Korean patients diagnosed with axSpA. They further compared the ASAS HI between patients with radiographic axSpA and patient with non-radiographic axSpA and identified predictors of the ASAS HI. Cross-sectional analyses were performed. In general, the data is interesting however the number of words could be decreased and the results could be presented more clearly.
Major
Title: The authors describe that they would like to compare the ASAS HI between AS patients and non-radiographic axSpA patients. In addition they also would like to identify predictors of the ASAS HI. However, the title only describes the second goal and not the primary goal. Please adjust the title.
The radiographs were assessed by a rheumatologist. Why were the radiographs assessed by a rheumatologist only and not also by a radiologist? Was this rheumatologist trained in assessing the radiographs?
Please define the predictors that were chosen for linear regression analysis and the reason for these predictors in the methods.
The order of the Tables and Figure is quite confusing. The results are better to understand if the authors would present Table 1 first, thereafter Figure 1, Table 2 etc. Please use the same order as the order of the text in the result section.
Table 1 describes not only the baseline characteristics but also other results. Please describe only the baseline characteristics in Table 1 and move the other results to another Table.
Figure 1 has a low image quality and is unreadable. Please increase the quality of the image and ensure that the text of the Figure could be read.
What is the clinical relevance of the differences between AS and non-radiographic patients and how must the readers interpret the level of the total ASAS HI score? And what is the clinical relevance of the predictors of the ASAS HI? Kiltz and colleagues (2018) described how results of the ASAS HI could be interpret. How could a mean of 3.8 for the ASAS HI be interpret and what does a difference of 1.1 between AS and non-radiographic axSpA say? And is a beta of -1.045 for current alcohol use in all patients clinically relevant? What does it tell us?
Could Table 3, 4 and 5 be merged together in one Table? It would make it easier to compare between the different subgroups.
The authors are inconsistent in reporting their results. In Table 3 male gender and HLA-B27 positivity are added to the multivariable analysis, while the factors do not show a p-value<0.20 in the univariable analysis and furthermore this is not done in Table 5. The authors should also mention in the methods section that they add gender, age and HLA-B27 status to the multivariable analyses irrespective of the p-value. Please add the multivariable analysis to Table 5 or remove male gender and HLA-B27 positivity from the analysis presented in Table 3.
The primary aim of the authors is to compare the ASAS HI between AS patients and patients with non-radiographic axSpA and to report the frequencies of the different items of the ASAS HI. However, in the discussion these outcome are hardly discussed nor compared to results in the literature. More attention should be paid to this primary aim in the discussion.
The conclusion of the abstract is well described, however the conclusion of the discussion is not a conclusion but a description of the goals that the authors aimed to answer. Please add which predictors were found and which ASAS HI items.
Minor
Did the patients received a diagnosis of axSpA before the axSpA classification criteria were applied? Classification criteria could only be applied when the patients already received a diagnosis.
The conclusion of the abstract that ASAS HI is well correlated with several disease parameters comes unexpectedly. The authors could give an example in the results section of the abstract of those disease parameters.
The objectives of the introduction could be defined a bit more clearly. It could be mentioned that the ASAS HI is compared in AS patients vs. non-radiographic patients and that the authors would like to identify predictors.
What is the sacroiliitis grade at baseline? Please report this in the results. It would also be more interesting to provide the total sacroiliitis grade instead of the averaged sacroiliitis grade. This would show what would happen with the ASAS HI if a patients has a higher sacroiliitis grade.
Why was only current smoking and consuming alcohol reported and not also former smokers or users of alcohol?
Please mention the purpose of using spearman correlation coefficients for ASAS HI and PROs. For example, “Spearman correlation coefficients were calculated to assess the correlation between..”.
It would be easier for the readers to understand the results of the most common items of the ASAS HI if the authors would write first the description of the item, thereafter the number of the item. Most readers will not know which item represents which domain. For example: “The most frequent item that patients experienced was pain (Item no. 1, 57% in AS, 44.8% in nr-axSpA”.
Not all SpA features are described in the baseline Table, for example if patients were suffering from inflammatory back pain. Could you please add these features?
Table 1 contains too much information and therefore it does not provide a clear overview. Both categories of dichotomous variables are described. It would be better to provide only one category instead of both.
Was the ASAS HI calculated if one or more items were missing? How did the authors handle missing values?
Result section 3.2: The authors write: “The absolute value of the Spearman correlation coefficient was highest between ASAS HI and…”. The results could be presented more clearly by telling what the results mean as this guides the reader through the results. For example: “Quality of life and functioning were highly correlated with the ASAS HI (EQ-6D-TTO Rho=… etc. and BASFI rho=..etc”.
The direction of the association or correlation etc. is sometimes missing. Please provide if an association/correlation is positive or negative.
Results section 3.3: It is not clear which patient group is assessed in the first sentences of this paragraph. I assume that all axSpA patients are investigated. Please mention the patient group. Further, please write down what is different about HLA-B27 status between the analysis in all patients and the analysis in the AS subgroup.
Could the authors think of a reason or hypothesis why patients who consumed alcohol had lower disease outcomes? Please add this to the discussion.
To my opinion is a study population of 372 patients not a limitation as mentioned in the discussion. The group of non-radiographic axSpA is somewhat small.
Author Response
Response to Reviewer 1 Comments
The authors described the ASAS health index in cohort of Korean patients diagnosed with axSpA. They further compared the ASAS HI between patients with radiographic axSpA and patient with non-radiographic axSpA and identified predictors of the ASAS HI. Cross-sectional analyses were performed. In general, the data is interesting however the number of words could be decreased and the results could be presented more clearly.
Major
Point 1: Title: The authors describe that they would like to compare the ASAS HI between AS patients and non-radiographic axSpA patients. In addition they also would like to identify predictors of the ASAS HI. However, the title only describes the second goal and not the primary goal. Please adjust the title.
Response 1 : We thank the referee for the comment about the title. We revised the title as including the primary goal of the present study. We highlighted revised title by using red colored text.
Revised Title : Predictors of ASAS Health Index in Axial Spondyloarthritis and comparison of ASAS health index between ankylosing spondylitis and non-radiographic axial spondyloarthritis: Data from the Catholic Axial Spondyloarthritis COhort (CASCO)
Point 2: The radiographs were assessed by a rheumatologist. Why were the radiographs assessed by a rheumatologist only and not also by a radiologist? Was this rheumatologist trained in assessing the radiographs?
Response 2: We absolutely understand the reviewer’s concern. However, thee rheumatologist, Min, who assessed the radiographic scoring in the present study, has been highly trained for scoring the sacroiliitis and mSASSS. As a result, he is responsible for measuring of radiologic scoring of axSpA even in global clinical trial of IL-17 inhibitor.
Point 3 : Please define the predictors that were chosen for linear regression analysis and the reason for these predictors in the methods.
Response 3: The predictors chosen for linear regression model were selected according to the previous studies which investigated predictors of Health related QoL and structural damage in spondyloarthritis, because predictors of ASAS HI is not known yet. We added the reason of choosing candidate variables included in regression analysis in method section. (Line 122-124)
Referrence :
1) Law L et al. Factors related to health-related quality of life in ankylosing spondylitis, overall and stratified by sex. Arthritis Res Ther. 2018 Dec 27;20(1):284.
2) Molnar C et al. TNF blockers inhibit spinal radiographic progression in ankylosing spondylitis by reducing disease activity: results from the Swiss Clinical Quality Management cohort. Ann Rheum Dis. 2018 Jan;77(1):63-69.
Point 4: The order of the Tables and Figure is quite confusing. The results are better to understand if the authors would present Table 1 first, thereafter Figure 1, Table 2 etc. Please use the same order as the order of the text in the result section.
Response 4: As the reviewer’s comment, we rearranged the Tables and Figures. (Table 1 -> Table 2 -> Figure 1 -> Table 3 -> revised Table 4(we merged the Table 3 to 5 into revised Table 3, as reviewer recommended in point 7))
Point 5: Table 1 describes not only the baseline characteristics but also other results. Please describe only the baseline characteristics in Table 1 and move the other results to another Table.
Figure 1 has a low image quality and is unreadable. Please increase the quality of the image and ensure that the text of the Figure could be read.
Response 5: We apologize for low image quality of figure 1. And we enhanced the quality of figure 1 which could identify the texture of figure 1. The information included in Table 1 is baseline characteristics of the enrolled patients when they were enrolled at CASCO. As reviewer’s comment, we separated the other results than baseline characteristics to Table 2.
Point 6: What is the clinical relevance of the differences between AS and non-radiographic patients and how must the readers interpret the level of the total ASAS HI score? And what is the clinical relevance of the predictors of the ASAS HI? Kiltz and colleagues (2018) described how results of the ASAS HI could be interpret. How could a mean of 3.8 for the ASAS HI be interpret and what does a difference of 1.1 between AS and non-radiographic axSpA say? And is a beta of -1.045 for current alcohol use in all patients clinically relevant? What does it tell us?
Response 6: As the referee commented, thee clinical relevance of ASAS HI in present study is limited. That’s because this included only baseline comparison. Although, in present study, the baseline difference of ASAS HI between AS and nr-axSpA could only be interpreted as AS patients are in inferior health state than nr-axSpA, but ASAS HI is worthy in that ASAS HI could approach patient’s disability, health, and functioning systematically. Furthermore, as mentioned in discussion section (Line 207-208), future studies could include ASAS HI as treatment response tool, and offer more information of treatment response, especially how actually patients feel. In addition, Kiltz and colleague (Kiltz, U., et al. (2018). Measurement properties of the ASAS Health Index: results of a global study in patients with axial and peripheral spondyloarthritis. Ann Rheum Dis 77(9): 1311-1317.) offerred that treatment response could be measured by calculating standardised response mean (SRM) of ASAS HI in each medication.
In multivariable linear regression analysis, current alcohol drinking showed β of -1.045, and this means that patients who drink alcohol currently have lower ASAS HI score than non-drinker for about 1 point. Aforementioned statement is applicable to other predictors such as economic status and mSASSS. Results of present study has limited usefulness due to cross-sectional study design, but in future studies which assess ASAS HI as one of the treatment response parameter, researcher should consider that patient’s baseline characteristics including drinking status, economic status, and structural damage (mSASSS) could act as confounding factor of ASAS HI.
Point 7: Could Table 3, 4 and 5 be merged together in one Table? It would make it easier to compare between the different subgroups.
Response 7: We thank the referee for helpful suggestion. As the referee commented, we merged the results of Table 3 to 5 into new Table 4. The title of revised Table 4 is marked by red color. (Line 185)
Point 8: The authors are inconsistent in reporting their results. In Table 3 male gender and HLA-B27 positivity are added to the multivariable analysis, while the factors do not show a p-value<0.20 in the univariable analysis and furthermore this is not done in Table 5. The authors should also mention in the methods section that they add gender, age and HLA-B27 status to the multivariable analyses irrespective of the p-value. Please add the multivariable analysis to Table 5 or remove male gender and HLA-B27 positivity from the analysis presented in Table 3.
Response 8: In table 3 of the original manuuscript, P value of male gender and HLA-B27 were 0.095, and 0.090, which are both under P value 0.20. In table 5, alcohol drinking only showed P value under 0.20, therefore multivariable linear regression analysis was not available.
Point 9: The primary aim of the authors is to compare the ASAS HI between AS patients and patients with non-radiographic axSpA and to report the frequencies of the different items of the ASAS HI. However, in the discussion these outcome are hardly discussed nor compared to results in the literature. More attention should be paid to this primary aim in the discussion.
The conclusion of the abstract is well described, however the conclusion of the discussion is not a conclusion but a description of the goals that the authors aimed to answer. Please add which predictors were found and which ASAS HI items.
Response 9: In result and discussion section, we stated the most frequently checked items and discussed that items proposed by AS patients (item 1 and 5) were top 2 items checked as “I agree” by axSpA patients. (Line 143-145, 202-205) It was interesting that items proposed by AS patients were most commonly checked as “I agree”, and we mentioned that aforementioned result support ASAS HI could assess patient’s actual discomfort in daily life.
We agree that we did not stated the most frequently checked areas which patients feel discomfort, and interpretation of this in conclusion section of abstract. We added the statement at the conclusion section of abstract. (Marked by red color, Line 31-33) The predictors of ASAS HI found in present study are mentioned in conclusion section. (Line 35-37) We highlighted changes within the document using red colored text.
Minor
Point 10: Did the patients received a diagnosis of axSpA before the axSpA classification criteria were applied? Classification criteria could only be applied when the patients already received a diagnosis.
Response 10: As shown in Table 1, mean disease duration of enrolled patients were 7.6 years. In some of the patients, who were diagnosed as AS before ASAS classification criteria was made, diagnosis was made according to modified New York criteria. The definition and purpose of diagnostic criteria and classification criteria are different. However, many rheumatologic disorders are decided by classification criteria nowadays. In addition, the present study was a result from prospective cohort study, applying classification criteria might be more suitable than diagnostic criteria.
Reference
1) Aggarwal R. et al. Distinctions between diagnostic and classification criteria? Arthritis Care Res (Hoboken). 2015 Jul;67(7):891-7.
Point 11: The conclusion of the abstract that ASAS HI is well correlated with several disease parameters comes unexpectedly. The authors could give an example in the results section of the abstract of those disease parameters.
Response 11: We thank the referee for the helpful comment to strengthen the manuscript. As the referee indicated, we added the example of the SpA related parameters in the results section of the abstract. The revised sentence was marked by red colored text. (Line 25)
Point 12: The objectives of the introduction could be defined a bit more clearly. It could be mentioned that the ASAS HI is compared in AS patients vs. non-radiographic patients and that the authors would like to identify predictors.
Response 12: As reviewer’s recommend, we rewrite the objectives of the present study in last paragraph of introduction. (Marked as red color, Line 61-63)
Point 13: What is the sacroiliitis grade at baseline? Please report this in the results. It would also be more interesting to provide the total sacroiliitis grade instead of the averaged sacroiliitis grade. This would show what would happen with the ASAS HI if a patients has a higher sacroiliitis grade.
Response 13: Grade of sacroiliitis at baseline meant the grade of sacroiliitis measured when the patients were enrolled in CASCO. To enhance the readers’ comprehension, we stated the grade of sacroiliitis was measured at the time of enrolment to CASCO. (Line 89-90) And we added the information of grade of sacroiliitis in Table 1. As reviewer’s comment, we performed linear regression analysis by including total grade of sacroiliitis (sum of right and left sacroiliitis grade), however the result did not show significant change. (total axSpA univariable beta = 0.255, SE = 0.082, P=0.002, AS univariable beta 0.247, SE=0.155, P=0.113, nr-axSpA univariable beta = -0.040, SE = 0.302, P=0.895) We revised the Table 4 by including total sacroiliitis grade instead of average grade of sacroiliitis. (revised Table 4)
Point 14: Why was only current smoking and consuming alcohol reported and not also former smokers or users of alcohol?
Response 14: We agree that former smoking / former drinking is also important and could affect long term disease parameter or structural damage of spine. But we hypothesized that current status might be more important in current state of ASAS HI because it represent current global health status of axSpA patient. As commented in ASAS HI, all items should be checked according to how patient’s feel at this moment.
ASAS HI
Point 15: Please mention the purpose of using spearman correlation coefficients for ASAS HI and PROs. For example, “Spearman correlation coefficients were calculated to assess the correlation between..”.
Response 15: As the reviewer’s comment, we added the purpose of using spearman correlation coefficients for ASAS HI and PROs. Revised sentence are marked by red color. (Line 121-122)
Point 16: It would be easier for the readers to understand the results of the most common items of the ASAS HI if the authors would write first the description of the item, thereafter the number of the item. Most readers will not know which item represents which domain. For example: “The most frequent item that patients experienced was pain (Item no. 1, 57% in AS, 44.8% in nr-axSpA”.
Response 16: We thank the referee for the important comment to strengthen the manuscript. Describing all items in the manuscript might take up a large portion of the main manuscript, so we added the supplementary data which shows whole items of ASAS HI. (Line 145-146, 274)
Point 17: Not all SpA features are described in the baseline Table, for example if patients were suffering from inflammatory back pain. Could you please add these features?
Response 17: Inflammatory back pain is mandatory feature when using ASAS classification criteria for axSpA. Therefore, we only included patients who suffered inflammatory back pain and eventually classified as axSpA in CASCO. All patients had inflammatory back at the time of diagnosis, but the current status of inflammatory back pain was not checked at the time of enrolment. Furthermore, some SpA features such as family history of SpA, good response to NSAID were not checked at the time of enrolment.
Point 18: Table 1 contains too much information and therefore it does not provide a clear overview. Both categories of dichotomous variables are described. It would be better to provide only one category instead of both.
Response 18: We think the referee commented an important issue. As the reviewer’s comment, we erased some of the variables which displays similar information, such as ESR, CRP, ASAS NSAID index, and syndesmophyte count in the revised manuscript.
Point 19: Was the ASAS HI calculated if one or more items were missing? How did the authors handle missing values?
Response 19: Thank you for pointing out important issue. Some items are not applicable or patient refuse to answer. In this case, the ASAS HI was calculated by (item summation score/17-number of missing items) * 17. In case of missing items exceed more than 20% of total items, the ASAS HI was not calculated and excluded from the analysis. We added this information on the method section by red colored text. (Line 101-103)
Point 20: Result section 3.2: The authors write: “The absolute value of the Spearman correlation coefficient was highest between ASAS HI and…”. The results could be presented more clearly by telling what the results mean as this guides the reader through the results. For example: “Quality of life and functioning were highly correlated with the ASAS HI (EQ-6D-TTO Rho=… etc. and BASFI rho=..etc”.
Response 20: As reviewer’s comment, we revised the sentence to enhance the reader’s comprehension. Revised sentence is colored by red. (Line 168-169)
Point 21: The direction of the association or correlation etc. is sometimes missing. Please provide if an association/correlation is positive or negative.
Response 21: We apologize for missing of direction of correlation / association. As the referee indicated, we added the direction “positive” or “negative” on the manuscript. (Marked as red color, Line 170, 183)
Point 22: Results section 3.3: It is not clear which patient group is assessed in the first sentences of this paragraph. I assume that all axSpA patients are investigated. Please mention the patient group. Further, please write down what is different about HLA-B27 status between the analysis in all patients and the analysis in the AS subgroup.
Response 22: We thank the referee for the important comment. We revised the first sentence of result 3.3 to clarify the group included in first linear regression analysis (Marked by red color, Line 174-175). HLA-B27 positivity was higher in AS group than nr-axSpA. The discrepant results of HLA-B27 status between total axSpA and AS might arise from the lower positivity of HLA-B27 in total axSpA group. As reviewer’s comment, we added discussion of aforementioned issue on discussion section. (Marked by red color, Line 257-261)
Point 23: Could the authors think of a reason or hypothesis why patients who consumed alcohol had lower disease outcomes? Please add this to the discussion.
Response 23: As reviewer’s comment, the reason for why alcohol drinker had negative association with ASAS HI is not clear. We discussed about this on discussion section already. (Line 229-234) More information and longitudinal follow up data are needed to clarify the effect of alcohol on ASAS HI. We did not add hypothesis or reason because adding hypothesis might be perceived to readers as over-interpretation.
Point 24: To my opinion is a study population of 372 patients not a limitation as mentioned in the discussion. The group of non-radiographic axSpA is somewhat small.
Response 24: We absolutely agree with the reviewer’s comment. We revised the limitation section which pointed out relative small sample size of nr-axSpA. Revised sentence are marked by red color. (Line 262-263) Thank you for your comments.

Reviewer 2 Report
This is a solid performed study on ASA-HI in an axspa group consisting of both radiologic and non radiologic ax-spa. The outcome are interesting but not spectacular. I have no major rpoblems with the paper.
In their introduction they talk about the effect of damage in asa-hi. However in their analysis it is just one of the parameters. I would have preferred a paper more focussed ont his aspect. Especially cause this is often a problem in clinical care. Als the fact taht mssasis not a predictor in nr-axpsa, as the name allready states the absence of radiologic damage. That needs to be changed in their discussion.
I am also a bit worried about the number of predictors they are testing. this could lead to finding several predictors by chance.
Author Response
Response to Reviewer 2 Comments
This is a solid performed study on ASA-HI in an axSpA group consisting of both radiologic and non radiologic axSpA. The outcome are interesting but not spectacular. I have no major problems with the paper.
Point 1: In their introduction they talk about the effect of damage in asas-hi. However in their analysis it is just one of the parameters. I would have preferred a paper more focussed on this aspect. Especially cause this is often a problem in clinical care. Also the fact that msasss not a predictor in nr-axpsa, as the name allready states the absence of radiologic damage. That needs to be changed in their discussion.
Response 1: We agree with the reviewer’s comment, so we erased the phrase which mentioned structural damage in introduction section. (Marked by red color, Line 59)
Actually, nr-axSpA does not mean that nr-axSpA patients do not have structural damage on spine or sacroiliac joint. Among axSpA patients who do not satisfy modified New York criteria (for example, right side sacroiliitis grade 1 & left side sacroiliitis grade 2), but has features of axSpA such as active inflammatory lesion on MRI, are classified as nr-axSpA. Furthermore, in some case, structural damage solely present in spine, but not in sacroiliac joint. Therefore mSASSS can be measured in nr-axSpA.
Reference
1) Huang J et al. Discriminating Value of Calprotectin in Disease Activity and Progression of Nonradiographic Axial Spondyloarthritis and Ankylosing Spondylitis. Dis Markers. 2017;2017:7574147.
2) Lorenzin M et al. Biomarkers, imaging and disease activity indices in patients with early axial spondyloarthritis: the Italian arm of the SpondyloArthritis-Caught-Early (SPACE) Study. Reumatismo. 2017 Aug 3;69(2):65-74.
3) Ez-Zaitouni Z et al. The yield of a positive MRI of the spine as imaging criterion in the ASAS classification criteria for axial spondyloarthritis: results from the SPACE and DESIR cohorts. Ann Rheum Dis. 2017 Oct;76(10):1731-1736.
Point 2: I am also a bit worried about the number of predictors they are testing. this could lead to finding several predictors by chance.
Response 2: We understand the reviewer’s comment. However, in multivariable regression analysis of total axSpA and AS, only seven variables were included as predictors. Number of patients (total axSpA (N=357), AS (N=261)) might be sufficient to included seven variables in multivariable linear regression analysis.

Reviewer 3 Report
Minor comments :
- in the abstract : the authors should provide the number of patients
- design of the study : why did the authors not include a control group (mechanical low back pain for example) ? Please comment
- results : in table 4 it appears that HLA-B27 positivity is negatively associated ASAS HI in univariable and multivariable regression analysis, but htis does not appear in the discussion. Please comment on that.
Author Response
Response to Reviewer 3 Comments
Minor comments :
Point 1: - in the abstract : the authors should provide the number of patients
Response 1: We thank for the comment. We added the total number of the patients in result section of abstract. (Marked by red color, Line 21-22)
Point 2: design of the study : why did the authors not include a control group (mechanical low back pain for example) ? Please comment
Response 2: As reviewer’s comment, including control (such as mechanical low back pain patients) would be interesting. However, the CASCO only included patients diagnosed as axSpA. Furthermore, ASAS HI was validated in axSpA and peripheral SpA, but not in mechanical back pain. Therefore, applying ASAS HI in control group was inappropriate.
Point 3: - results : in table 4 it appears that HLA-B27 positivity is negatively associated ASAS HI in univariable and multivariable regression analysis, but this does not appear in the discussion. Please comment on that.
Response 3: The association between HLA-B27 positivity and clinical parameters associated with SpA are rarely studied. One study showed HLA-B27 negative AS patients had higher BASDADI and BASFI score than HLA-B27 positive AS patients. Although clinical parameters of previous study (2018 Arevalo. M et al, Arthritis Res Ther 20(1): 221.) and our study were different, and previous study did not perform regression analysis, the results of two study consistently suggest possible negative association between HLA-B27 positivity and clinical parameters. We added this point on discussion section. (Marked by red color, Line 252-257)
Referrence
1) Arevalo, M., et al. (2018). "Influence of HLA-B27 on the Ankylosing Spondylitis phenotype: results from the REGISPONSER database." Arthritis Res Ther 20(1): 221.

Round 2
Reviewer 1 Report
The authors have addressed almost all my questions and remarks. The quality of Figure 1 is good and readable. I would like to get back to:
Point 16: I would like to thank the authors for adding the ASAS HI as supplementary data. I understand that describing all items of the ASAS HI could take up a large portion of the manuscript. However, readability of the results section would improve if the authors could add the description of all items of the ASAS HI mentioned in the text of the result section just once. So, if an item of the ASAS HI is mentioned for the first time in the result section, the authors could add the description to that item. When that item is mentioned for the second time, it is not necessary to give the description again. In this manner, it will not costs many words as only item no. 1, 3 and 5 are mentioned in the result section.
Point 17: Inflammatory back pain is not a mandatory feature in the ASAS classification criteria for axSpA, but chronic back pain is. Please see Rudwaleit et al (Ann Rheum Dis 2009;68:777-783). It is no problem that all patients had inflammatory back pain, but this needs to be stated.
Author Response
Point 16: I would like to thank the authors for adding the ASAS HI as supplementary data. I understand that describing all items of the ASAS HI could take up a large portion of the manuscript. However, readability of the results section would improve if the authors could add the description of all items of the ASAS HI mentioned in the text of the result section just once. So, if an item of the ASAS HI is mentioned for the first time in the result section, the authors could add the description to that item. When that item is mentioned for the second time, it is not necessary to give the description again. In this manner, it will not costs many words as only item no. 1, 3 and 5 are mentioned in the result section.
Response 16 : We thank the referee for the important comment to strengthen the manuscript. We added the description of item 1, 3, 5 on the manuscript. (Marked by blue color, Line 144-146)
Point 17: Inflammatory back pain is not a mandatory feature in the ASAS classification criteria for axSpA, but chronic back pain is. Please see Rudwaleit et al (Ann Rheum Dis 2009;68:777-783). It is no problem that all patients had inflammatory back pain, but this needs to be stated.
Response 17: As reviewer’s comment, we added the statement that all enrolled patients had inflammatory back pain. (Marked by blue color, Line 135)
